# Invasive and Noninvasive Nonfunctioning Gonadotroph Pituitary Tumors Differ in DNA Methylation Level of LINE-1 Repetitive Elements

**DOI:** 10.3390/jcm10040560

**Published:** 2021-02-03

**Authors:** Natalia Rusetska, Paulina Kober, Sylwia Katarzyna Król, Joanna Boresowicz, Maria Maksymowicz, Jacek Kunicki, Wiesław Bonicki, Mateusz Bujko

**Affiliations:** 1Department of Molecular and Translational Oncology, Maria Sklodowska-Curie National Research Institute of Oncology, 02-781 Warsaw, Poland; natarusetska@gmail.com (N.R.); paulina.kober@gmail.com (P.K.); sylwia_krol15@wp.pl (S.K.K.); joanna.boresowicz@gmail.com (J.B.); 2Department of Pathology and Laboratory Diagnostics, Maria Sklodowska-Curie National Research Institute of Oncology, 02-781 Warsaw, Poland; Maria.Maksymowicz@pib-nio.pl; 3Department of Neurosurgery, Maria Sklodowska-Curie National Research Institute of Oncology, 02-781 Warsaw, Poland; jkunickii@gmail.com (J.K.); neurochirurgia@coi.waw.pl (W.B.)

**Keywords:** DNA methylation, LINE-1, L1-ORF1p, pituitary tumor, gonadotropinoma, invasive growth

## Abstract

Purpose: Epigenetic dysregulation plays a role in pituitary tumor pathogenesis. Some differences in DNA methylation were observed between invasive and noninvasive nonfunctioning gonadotroph tumors. This study sought to determine the role of DNA methylation changes in repetitive LINE-1 elements in nonfunctioning gonadotroph pituitary tumors. Methods: We investigated LINE-1 methylation levels in 80 tumors and normal pituitary glands with bisulfite-pyrosequencing. Expression of two LINE-1 open reading frames (*L1-ORF1* and *L1-ORF2*) was analyzed with qRT-*PCR* in tumor samples and mouse gonadotroph pituitary cells treated with DNA methyltransferase inhibitor. Immunohistochemical staining against L1-ORF1p was also performed in normal pituitary glands and tumors. Results: Hypomethylation of LINE-1 was observed in pituitary tumors. Tumors characterized by invasive growth revealed lower LINE-1 methylation level than noninvasive ones. LINE-1 methylation correlated with overall DNA methylation assessed with HM450K arrays and negatively correlated with *L1-ORF1* and *L1-ORF2* expression. Treatment of αT3-1 gonadotroph cells with 5-Azacytidine clearly increased the level of *L1-ORF1* and *L1-ORF2* mRNA; however, its effect on LβT2 cells was less pronounced. Immunoreactivity against L1-ORF1p was higher in tumors than normal tissue. No difference in L1-ORF1p expression was observed in invasive and noninvasive tumors. Conclusion: Hypomethylation of LINE-1 is related to invasive growth and influences transcriptional activity of transposable elements.

## 1. Introduction

Pituitary neuroendocrine tumors (PitNETs) are frequently diagnosed intracranial tumors originating from different functional pituitary cells. Most of these tumors present with symptoms of particular pituitary hormone hypersecretion but a large proportion of pituitary tumors are nonfunctioning and develop without endocrinological syndromes [1]. The majority of nonfunctioning PitNETs originate from gonadotropic cells and are diagnosed by immunohistochemical staining with antibodies against follicle-stimulating hormone (FSH), luteinizing hormone (LH), α-subunit and SF1 (steroidogenic factor 1) transcription factor. They are commonly considered benign, well-delineated neoplasms; however, a notable percentage of these nonfunctioning tumors invade the adjacent structures, causing important clinical implications. Invasive growth hampers complete tumor resection, which is a basic treatment of nonfunctioning gonadotroph tumors and results in tumor recurrence [2,3,4]. It is also one of the most important prognostic features for patients suffering from PitNETs [4]. Identification of biological determinants of aggressive growth of pituitary tumors could provide clinical benefit in regard to patient prognosis, adjuvant treatment and recommendation of more frequent follow-up visits for some patients [5]. Importantly, analysis of invasive and noninvasive nonfunctioning gonadotroph PitNETs revealed some differences in DNA methylation patterns [6,7,8].

In human cancer, both local and global DNA methylation changes occur in the genome [9]. A decrease in the overall DNA methylation level was observed in many solid tumors, contributing to genome instability [9] and elevated mutation rates [10]. This tumor-specific global DNA hypomethylation includes decreased methylation at repetitive DNA sequences, which allows for the activation of the transposable elements (TEs) [11]. There are two main classes of TEs in the human genome: retrotransposons and DNA transposons [12]. The only autonomous, active TE in the human genome is Long Interspersed Nuclear Element-1 (LINE-1), which belongs to retrotransposons [12]. LINE-1 contains two open reading frames, *L1-ORF1* and *L1-ORF2*, that encode for L1-ORF1p (RNA binding protein) and L1-ORF2p (reverse transcriptase and endonuclease) proteins, respectively. These two proteins cooperate to introduce the retrotransposon sequence into new chromatin target sites [13].

LINE-1 sequences constitute up to 17% of the human genome and are generally methylated, which keeps them silent and repressed. However, following global reduction in methyl-CpG content in cancer, LINE-1 sequences become demethylated [13].

Reduced LINE-1 methylation was reported in many tumor types and was found to be related to clinicopathological features and patients’ prognosis [14,15,16,17,18,19,20,21].

In this study, we determined the methylation level of LINE-1 repetitive elements in PitNETs and made an attempt to evaluate its role in invasive tumor growth.

## 2. Experimental Section

### 2.1. Patients and Samples

Tumor fragments from 80 patients were collected in years 2010–2016 during transsphenoidal surgery and immediately frozen in liquid nitrogen, for storage at −80 °C. Each tumor was assessed histopathologically with immunohistochemistry and ultrastructural investigation with electron microscopy. All the samples were nonfunctioning gonadotroph PitNETs. Sixty-eight samples were newly diagnosed, while 12 samples were recurrent tumors. Median follow-up of 71 months (range: 50–122 months) was available for the patients. Ten patients experienced tumor recurrence during this follow-up. Patients characteristics are summarized in Table 1; detailed patients’ clinical data are presented in Appendix A.

Tumor invasion and main directions of invasive growth were evaluated on the pre-operative MRI. Cavernous sinus invasion was defined by extension of tumor beyond the line corresponding to the lateral tangents of the intracavernous carotid artery (Grade 3 and 4), as defined by Knosp et al. [22,23]. Infrasellar direction of invasive growth was considered if bone and dura of the sellar floor, sphenoid sinus and clivus were invaded. The suprasellar expansion was considered invasive only if leptomeningeal infiltration was observed. Invasiveness of pituitary tumors was taken into account when it was confirmed by the surgeon with endoscopic inspection of infiltrated areas and/or by histology.

Approval was obtained from the local ethics committee in Maria Sklodowska-Curie National Research Institute of Oncology in Warsaw for experimenting on human samples. Each patient provided informed consent for using tissue samples for scientific purposes.

Five samples of normal human pituitary tissue from autopsies were used for DNA methylation analysis. No incidental pituitary tumors were identified in these tissue samples upon hematoxylin/eosin staining and histopathological evaluation. Postmortem interval (PMI) for the samples was in the range of 25–46 h (median 35.5). According to previously reported data, this PMI does not affect DNA methylation assessment with bisulfite-based methods [5]. Three formalin-fixed, paraffin-embedded (FFPE) samples of histopathologically confirmed normal pituitaries obtained from resected Rathke’s cleft cyst were used for immunohistochemical staining. Details of normal pituitary samples are shown in Appendix A.

DNA and RNA were isolated using QIAamp DNA Mini Kit (Qiagen, Hilden, Germany) and RNeasy Mini Kit (Qiagen, Hilden, Germany), respectively. Their quantity was determined spectrophotometrically with NanoDrop 2000 (Thermo Fisher Scientific, Waltham, MA, USA). DNA and RNA samples were stored at −20 °C and −80 °C, respectively.

### 2.2. Analysis of LINE-1 DNA Methylation Level

DNA methylation level of LINE-1 was measured with pyrosequencing as previously [24]. Briefly, DNA was bisulfite-treated using EpiTect kit (Qiagen, Hilden, Germany). Polymerase chain reaction (PCR) was conducted in 30-μL mixture containing 1xPCR buffer, 2 mM MgCl_2_, 0.25 mM dNTPs, 0.2 μM each primer (sequences in Appendix A), 0.5 U of FastStart DNA Polymerase (Roche Applied Science, Penzberg, Germany) and 1 μL of bisulfite-treated DNA. Cycling conditions were as follows: 94 °C for 3 min, followed by 40 cycles of 30 s at 94 °C, 30 s at 52 °C and 30 s at 72 °C with final elongation 7 min at 72 °C. PCR products were purified and analyzed using PyroMark Q24 System (Qiagen, Hilden, Germany), according to manufacturer’s protocol. The average methylation level of CpGs within analyzed sequences was calculated for each sample.

### 2.3. Analysis of Genome-Wide Methylation

Previously generated results of genome-wide DNA methylation profiling of 18 PitNETs with Infinium HumanMethylation450 BeadChip (HM450K) (Illumina) were used (deposited at Gene Expression Omnibus; GSE115783). Data were analyzed with ChAMP data analysis pipeline as described [6]. Median β-values for all microarray probes that passed the quality filtration procedure and probes of particular categories, according to lluminaHumanMethylation450k.db library annotation, were calculated and used for the analysis.

### 2.4. Reverse Transcription and Quantitative PCR

Expression levels of two LINE-1 transcripts, *L1-ORF1* and *L1-ORF2*, were determined using qRT-PCR. Reverse transcription of 500 ng of RNA was performed using the Transcriptor First Strand cDNA Synthesis Kit (Roche Applied Science, Penzberg, Germany). Power SYBR Green PCR Master Mix (Thermo Fisher Scientific, Waltham, MA, USA) was used. PCRs were run in 5-mL volumes containing 2.25 pmol of each primer (sequences in Appendix A) on a 7900HT Fast Real-Time PCR System (Applied Biosystems, Foster City, CA, USA) in 384-well format. Standard curves based on the amplification of known concentrations of cDNA template were used for determining PCR efficiency and 2^-deltaCT^ method was used for calculating relative expression. *GAPDH* served as a reference gene based on previous validation [25]. Previously used PCR primers for human [26] and mouse-specific [27,28] qRT-PCR assays were used.

### 2.5. Cell Culture and Treatment

Two mouse gonadotroph cell lines LβT2 [29] and αT3-1 [30] were a kind gift from Dr Pamela Mellon (University of California, San Diego, CA, USA). The cells were cultured in Dulbecco’s modified Eagle’s medium with high glucose and pyruvate (DMEM GlutaMAX™, Gibco, Life Technologies, Paisley, UK), supplemented with 10% fetal bovine serum (FBS, Sigma-Aldrich, St. Louis, MO, USA) and antibiotics: 100 U/mL penicillin, 100 µg/mL streptomycin (HyClone™ Penicillin-Streptomycin Solution, GE Healthcare Life Sciences, Logan, UT, USA). The cell cultures were maintained at 37 °C in a humidified atmosphere of 5% CO_2_. Cells were mycoplasma-free according to PCR assay.

LβT2 and αT3-1 cells were seeded on 6-well plates at a density of 1.5 × 10^5^ or 0.85 × 10^5^ cells/well, respectively, and exposed to increasing concentrations of 5-Azacytidine (5-AzaC) (0.1, 0.5, 1, 5 and 10 μM) or DMSO (solvent control). The medium containing 5-AzaC was refreshed every 24 h. After 72 h of treatment, the cells were collected by scraping and total RNA was isolated using AllPrep^®^ DNA/RNA/miRNA Universal Kit (Qiagen, Hilden, Germany) according to the manufacturer’s protocol. The treatment of cells with 5-AzaC was performed on two different cell passages (two independent experiments).

### 2.6. Immunohistochemical Staining

Immunohistochemical staining (IHC) was performed on 4-μm FFPE tissue sections from 53 PitNETs and 3 normal pituitary samples using the Envision Detection System (DAKO, Glostrup, Denmark). Sections were deparaffinized with xylene and rehydrated in a series of ethanol solutions of decreasing concentration. Heat-induced epitope retrieval was carried out in a Target Retrieval Solution pH 9 (DAKO) in a 96 °C water bath, for 30 min. After cooling the retrieval solution for 30 min at room temperature, the slides were treated for 5 min with a Blocker of Endogenous Peroxidase (DAKO). Monoclonal antibody (MABC1152, Sigma-Aldrich, Saint Louis, MO, USA) was used to detect L1-ORF1p expression in tumor tissue samples. Overnight incubation at 4 °C with primary antibody (dilution 1:1000) was applied. The color reaction product was developed with 3,3′-diaminobenzidine tetrahydrochloride (Dako) as a substrate, and nuclear contrast was achieved with hematoxylin counterstaining. Analysis of immunohistochemical cytoplasmic reactivity was performed by evaluation of color reaction intensity (scored from 0 to 3: none, weak, moderate and strong expression). Quantification of immunoreactivity in each sample was also performed by H-score calculation using previously described formula [31]. Colorectal cancer and normal kidney tissue samples served as positive and negative controls, respectively, according to previously reported results of L1-ORF1p expression in various human tissues [32].

### 2.7. Statistical Analysis

Quantitative continuous variables were analyzed by a two-sided Mann–Whitney U test and Spearman’s rank correlation. Nonparametric Friedman test was used for comparing more than two groups of samples. Fisher’s exact test was used for analysis of proportions. Significance threshold α = 0.05 was adopted. Data were analyzed using GraphPad Prism (GraphPad Software, San Diego, CA, USA).

## 3. Results

### 3.1. LINE-1 DNA Methylation in Nonfunctioning Gonadotroph PitNETs and Normal Pituitary

LINE-1 DNA methylation levels were determined in five samples of normal pituitary gland and 80 nonfunctioning gonadotroph PitNETs, including 54 invasive and 26 noninvasive tumors. DNA methylation level in normal samples ranged from 75.05% to 80.9% (median 78.09%), while in PitNETs, it ranged between 37.48% and 77.36%, with a median of 71.21%. Comparison of tumor and normal sections showed significantly lower levels of DNA methylation in tumors (mean value 69.17% vs. 77.84%, respectively; *p* < 0.0001). When PitNETs stratified according to tumor invasion status were compared, lower LINE-1 methylation levels were observed in invasive PitNETs than in noninvasive ones (mean value 68.07% vs. 71.45%, respectively; *p* = 0.0192) (Figure 1A). We verified whether LINE-1 methylation level can serve as a classifier for stratifying the patients according to invasiveness status with receiver operating characteristic (ROC) curve analysis. Unfortunately, it showed a relatively low diagnostic value with area under the ROC Curve (AUC) = 0.663 (95% CI 0.5451 to 0.7802; *p* = 0.0139) (Figure 1B).

The relationship between LINE-1 methylation and additional clinical parameters related to aggressive PitNET growth was analyzed. LINE-1 methylation in tumors with proliferation index > 3% was compared with those with proliferation index < 3%, but no difference was found. Similarly, no difference was noted in comparison of newly diagnosed and recurrent tumors, in comparison of patients with tumor recurrence in follow-up and patients without evidence of recurrence, as well as in tumors smaller than 10 mm vs. larger than 10 mm. No difference was also observed when comparing PitNETs with and without ultrastructural features of oncocytic tumors.

### 3.2. LINE-1 and Overall DNA Methylation Determined with HM450K Microarrays

Eighteen of the tumor samples have been previously subjected to DNA methylation profiling with HM450K arrays (Illumina). We assessed whether LINE-1 methylation levels correlated with the overall genome-wide DNA methylation, estimated by median β-value of HM450K array probes. A significant correlation between LINE-1 methylation and overall β-value was observed (Spearman R = 0.579, *p* = 0.0118) (Figure 1C).

### 3.3. LINE-1 DNA Methylation and Expression of LINE-1 Open Reading Frame

Changes in DNA methylation may affect the expression of LINE-1 retrotransposon, which in turn plays a role in neoplastic transformation [13]. We determined the expression level of two open reading frames of LINE-1 retrotransposon (*L1-ORF1* and *L1-ORF2*) in tumor samples included in the LINE-1 methylation assessment. Unfortunately, when we compared the expression levels of *L1-ORF1* and *L1-ORF2* between invasive and noninvasive tumors, no significant difference was observed (Figure 2A).

Analysis of LINE-1 methylation and expression levels showed a significant negative correlation for both mRNAs: *L1-ORF1* (Spearman R = −0.2680; *p* = 0.0169) and *L1-ORF2* (Spearman R = −0.2720; *p* = 0.0153) (Figure 2B). For the additional evaluation of the relationship between DNA methylation and the expression level of LINE-1, we treated gonadotroph pituitary cells with 5-AzaC, an inhibitor of DNA methyltransferases. αT3-1 and LβT2 cell lines, originated from immature and mature normal mouse gonadotroph cells, were exposed to step-wise increasing concentrations of 5-AzaC for 72 h. Treatment of αT3-1 cells with increasing concentrations of 5-AzaC resulted in a proportional increase in the expression of both *L1-ORF1* and *L1-ORF2* transcripts (*p* = 0.0082 and *p* = 0.0092, respectively) (Figure 2C). However, such a clear relationship between DNA methylation and LINE-1 expression was not observed in LβT2 cells. Although higher expression of *L1-ORF1* was noticed in these cells (*p* = 0.0389), no influence of 5-AzaC on *L1*-*ORF2* was found (Figure 2C).

### 3.4. L1-ORF1p Expression in Nonfunctioning Gonadotroph Tumor Tissue and Normal Pituitary Glands

The expression of protein 1 encoded by the LINE-1 open reading frame was determined in 53 nonfunctioning gonadotroph PitNETs (including 34 invasive and 19 noninvasive tumors) as well as in three normal pituitary samples. Previously validated antibody against N-terminus of L1-ORF1p (clone 4H1) [32] was used for immunostaining. We observed mainly cytoplasmic reactivity. In normal pituitary tissue, it was scored as low in two samples and moderate in one sample. Tumor tissue showed moderate and strong immunoreactivity. None of the tumor samples was rated as negative or weak for protein expression. A homogeneous expression pattern with evenly distributed color reaction within each sample was detected. Weak nuclear staining was found only in a few samples with high cytoplasmic reactivity. Immunoreactivity was quantified by calculating the H-score for each sample. The comparison of the H-score in normal pituitary tissue and nonfunctioning gonadotroph PitNETs showed significantly lower L1-ORF1p expression in normal tissue (*p* = 0.0063) (Figure 3C). No significant difference between invasive and noninvasive tumors was found in the analysis of the proportion of moderately and highly stained tissue sections as well as in the H-score comparison (Figure 3C). The results of the immunoreactivity evaluation are presented in Table 2, while representative images of immunohistochemical staining showing expression of L1-ORF1p are presented in Figure 3.

We made an attempt to determine the relationship between LINE-1 DNA methylation level and L1-ORF1p expression. Unfortunately, no differences were observed between samples with high and moderate immunoreactivity when LINE-1 DNA methylation levels were analyzed in tumors stratified according to L1-ORF1p immunoreactivity. No correlation between LINE-1 DNA methylation level and immunoreactivity measured with H-score was also observed.

## 4. Discussion

The phenomenon of genome-wide loss of DNA methylation is a hallmark of human cancer and includes the decrease in DNA methylation level in repetitive genomic elements [9]. This phenomenon is observed early during tumor development [33] and is considered a pro-oncogenic event [34]. In general, DNA hypomethylation at repetitive genomic elements is commonly observed in tumor samples as compared to normal counterparts [17,19,20]. Similarly, we observed lower LINE-1 methylation levels in nonfunctioning gonadotroph PitNETs than in normal pituitary samples. However, it should be noted that the pituitary gland is composed of different functional cell types and normal gonadotrophs represent only part of the normal gland [35].

A decrease in LINE-1 methylation has an important clinical implication in human tumors: it was found to be a negative prognostic factor in tumors of the colon [15], stomach [16], prostate [17], brain [18] and esophagus [19]. It is associated with characteristics of aggressive growth in prostate [16], ovarian [20] and gastric [21] cancer. Pituitary tumors are generally considered benign neoplasms and surgical treatment allows for long-term remission. However, some tumors exhibit invasive growth, which is an important clinical problem [14,15].

Invasiveness of pituitary tumors is one of the most important prognostic features which is believed to be determined by tumor biology [5]. Our results show that invasive nonfunctioning gonadotroph PitNETs have lower LINE-1 methylation than noninvasive ones. These data are concordant with findings from other human cancers where lower LINE-1 methylation is related to aggressive growth and worse prognosis [14,15,16,17,18,19,20]. The molecular markers of nonfunctioning gonadotroph PitNET aggressive growth are needed as they could be useful in the clinical decision-making process [5]. Patients with rationally determined increased risk of recurrence could benefit from more individual treatment—more frequent follow-up visits or adjuvant therapy. Unfortunately, the difference in the methylation level of LINE-1 between invasive and noninvasive tumors is not considerable enough to provide a useful classifier of invasive pituitary tumors, as shown by our ROC curve analysis.

We also tried to determine the relationship between LINE-1 methylation and additional clinical parameters related to aggressive PitNET growth, including proliferation index, tumor recurrence status or tumor size category (microadenoma vs. macroadenoma). No significant relation was found; however, it should be noted that the value of these analyses is limited due to the low number of samples in particular groups in comparisons. The vast majority of samples in the study group were low proliferating tumors (Ki67 < 3%) and macroadenomas, which hampers the analysis of these parameters. Since most PitNETs are slow-growing tumors, the follow-up of median 71 months is probably too short to obtain sufficient information on patients’ outcomes. In our cohort, only 10 patients experienced tumor recurrence during this follow-up period.

DNA methylation is considered an important mechanism for TE silencing. Therefore, cancer-related DNA hypomethylation is associated with their increased transcriptional activity [13].

Recently published data on multi-omics profiling of different PitNETs showed a correlation between genome-wide average DNA methylation (determined with methylation microarrays) and expression of various TEs [36]. Accordingly, our data on nonfunctioning gonadotroph PitNETs show that global genome-wide methylation assessed with HM450K methylation arrays is correlated with the level of LINE-1 methylation. The relationship that we observed is not strong (Spearman R = 0.58) but it should be noted that genome-wide methylation assessed with methylation arrays may not reflect the true genomic methylation level but rather the overall methylation status of genomic regions covered by array probes.

We determined the expression level of *L1-ORF1* and *L1-ORF2* in nonfunctioning gonadotroph PitNET samples with qRT-PCR assay. The samples showed a notable dispersion of the expression level values for both transcripts and a slight, significant correlation was observed between LINE-1 methylation and *L1-ORF1*/*L1-ORF2* expression. This observation supports the hypothesis that hypomethylation contributes to LINE-1 activation in pituitary tumors. Consequently, an increase in both LINE-1 transcripts was observed in αT3-1 cells treated with DNA methylation inhibitor (5-AzaC). This indicates that inhibition of DNA methylation promotes activation of repetitive elements. A similar effect to that of 5-AzaC treatment on repetitive elements was also found in some human solid tumors [37,38].

The relationship between 5-AzaC concentration and activation of LINE-1 was less pronounced in the other gonadotroph cell line, LβT2, where a lack of a dose-dependent inhibitory effect of 5-AzaC on *L1-ORF2* level was found. However, some increase in *L1-ORF1* level in LβT2 cells was detected. The higher sensitivity of αT3-1 to DNA methyltransferases inhibitor may be caused by the difference in lineage differentiation stage of these two cell lines. αT3-1 are immature gonadotropic cells while LβT2 are nearly fully differentiated cells. Epigenetic regulation, including editing of the DNA methylation pattern, plays an important role in the differentiation of gonadotropic pituitary cells [39,40,41]. Immature gonadotropic cells have a higher expression level of Tet1 protein, which is directly involved in active DNA demethylation [41]. This difference was clearly observed in the comparison of αT3-1 and LβT2 cells [41]. Probably, experimental inhibition of DNA methyltransferases is more effective in αT3-1 cells because these cells have higher level of physiological DNA demethylation activity.

When we compared the expression levels of LINE-1 transcripts between invasive and noninvasive PitNETs, the difference did not reach statistical significance so we cannot conclude that these two groups generally differ in LINE-1 open reading frame expression.

The assessment of LINE-1 transcripts with qRT-PCR is technically feasible for the screening of a large number of samples as used previously [26,42] but is probably biased by the fact that LINE-1 sequences are incorporated in intronic regions of various genes and can undergo “pass through” transcription not related to the L1 promoter. The precise evaluation of the expression of repetitive elements is very challenging and requires sophisticated methods based on whole transcriptome analysis [43]. For this reason, we analyzed LINE-1 expression at protein level for comparison of invasive and noninvasive gonadotroph PitNETs in terms of the expression of L1-ORF1p. The expression of L1-ORF1p is generally detected in many human cancers [32,44] as well as in the normal human brain [45]. In agreement with previous results from immunohistochemical staining with anti-L1-ORF1p antibody (CH4.1 clone) [32,46], we also observed predominantly cytoplasmic expression in each analyzed tissue section. L1-ORF1p expression was determined in three normal pituitary samples and in tumor tissues. The comparison showed higher protein expression in tumors than in normal glands, which resembles the results of comparisons of other tumors and their normal tissue counterparts [32,44]. In nonfunctioning gonadotroph PitNETs, we detected moderate or high protein expression, with no difference between invasive and noninvasive tumors. We believe that two factors could influence this lack of difference as well as the lack of correlation between LINE-1 methylation and L1-ORF1p expression in our results. The first is the difficulty in the quantification of immunohistochemical data, which always relies on subjective observation. The second are natural differences in L1-ORF1p expression levels between individual patients, which are not a result of impaired methylation but reflect normal inter-individual diversity. Such differences were observed in a previous examination of the normal human brain [45]. We believe that this inter-individual diversity might be also the cause of the higher immunoreactivity of one of three normal pituitary samples in our study.

A hypothesis that, in pituitary tumors, global loss of methylation contributes to chromosomal instability through the activation of TEs was recently proposed [36]. It is based on two observations: an inverse correlation between genome-wide methylation and expression of TEs and an inverse correlation between average genomic methylation level and number of chromosomal aberrations in PitNETs [36]. Retrotransposition of LINE-1 elements plays an important role in the accumulation of genomic abnormalities in some human cancers [47]. However, alternative oncogenic mechanisms of LINE-1 activity were also proposed, including protein truncating mutations [48] or activation of telomerase [49].

Our data show that genome-wide DNA methylation is correlated with methylation of LINE-1 sequences, which are a particular type of TEs. Moreover, lower methylation of LINE-1 contributes to an increase in LINE-1 mRNA expression. We cannot confirm the direct influence of LINE-1 methylation on the expression of LINE-1 protein, required for the effective retrotransposition, since we did not observe a difference in methylation level between tumors with high and moderate immunoreactivity against L1-ORF1p. Thus, the functional consequences of LINE-1 hypomethylation in nonfunctioning gonadotroph PitNETs for their invasive growth remain elusive.

## 5. Conclusions

Hypomethylation of LINE-1 repetitive elements occurs in nonfunctioning gonadotroph pituitary tumors. Invasive PitNETs are characterized by decreased DNA methylation at LINE-1 repetitive elements as compared to noninvasive tumors. LINE-1 methylation is correlated with overall genome-wide methylation determined with HM450K microarrays in tumor samples and the lower methylation level corresponds to increased expression of LINE-1-encoded RNA. Expression of L1-ORF1p is higher in nonfunctioning gonadotroph pituitary tumors than in normal pituitary glands.

## Figures and Tables

**Figure 1 jcm-10-00560-f001:**
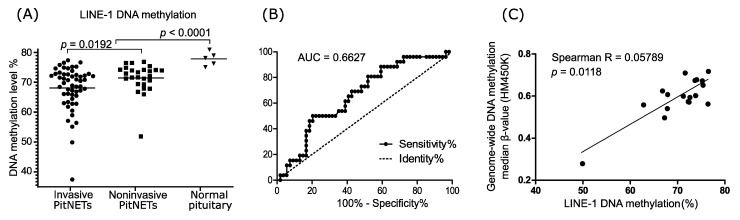
LINE-1 DNA methylation in nonfunctioning gonadotroph PitNETs. (**A**) Comparison of invasive (*n* = 54) and noninvasive (*n* = 26) tumors. Each dot represents average LINE-1 methylation level in the sample. Horizontal lines indicate mean values; (**B**) ROC (receiver operating characteristic) curve analysis for LINE-1 DNA methylation as classifier of invasive growth status; (**C**) Correlation between LINE-1 methylation and genome-wide methylation levels estimated as median β-value from HM450K arrays. Each dot represents PitNET sample (*n* = 18).

**Figure 2 jcm-10-00560-f002:**
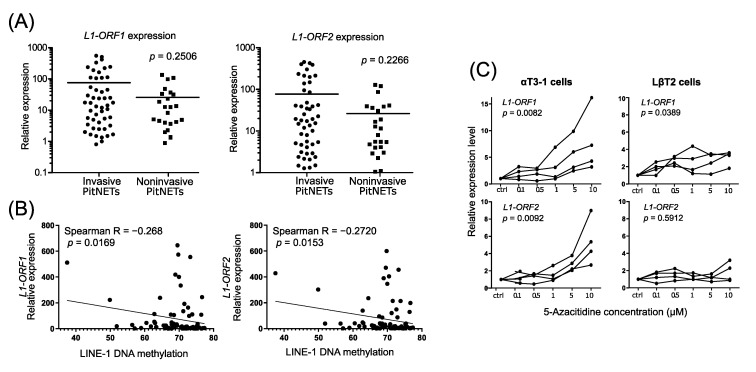
Expression of *L1-ORF1* and *L1*-*ORF2* transcripts. (**A**) Comparison of LINE-1 mRNA *L1-ORF1* and *L1-ORF2* levels in invasive (*n* = 54) and noninvasive (*n* = 26) nonfunctioning gonadotroph PitNETs. Each dot represents particular tumor sample, horizontal line indicates mean value; (**B**) Correlation analysis of LINE-1 DNA methylation end expression levels of *L1-ORF1* and *L1*-*ORF2.* (**C**) The expression of *L1-ORF1* and *L1*-*ORF2* in gonadotroph cell lines upon treatment with 5-AzaC. The results were normalized to control cells (CTRL). The lines connecting datapoints represent independent experiments.

**Figure 3 jcm-10-00560-f003:**
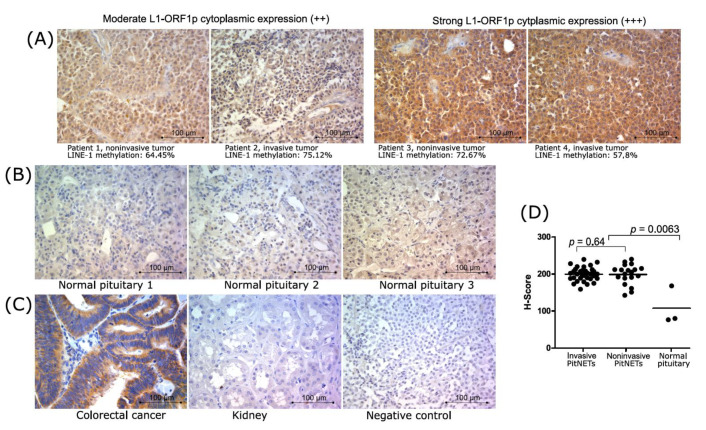
L1-ORF1p expression in nonfunctioning gonadotroph PitNETs. (**A**) Representative immunostaining showing expression of L1-ORF1p in tumor samples, magnification ×400. The results of LINE-1 DNA methylation assessment for each sample are provided below the tissue image; (**B**) Immunoreactivity against L1-ORF1p in normal pituitary sections, magnification ×400. (**C**) Results of immunostaining in colorectal cancer and normal kidney samples used as positive and negative controls, respectively, as well as technical negative control (procedure without primary antibody), magnifi-cation ×400; (**D**) The results of quantification of immunoreactivity in normal pituitary and tumor samples stratified according to invasive growth status with H-score calculation.

**Table 1 jcm-10-00560-t001:** Pituitary Tumor Patients’ Characteristics.

Nonfunctioning Gonadotroph PitNET Patients (Number of Patients)	80
**Demographical data**	
Age (years)	
Median; range	59.5; 34–82
Gender	
Male	46
Female	34
**Histopathology**	
Gonadotroph tumors	79
Plurihormonal (TSH, FSH, LH, α-subunit)	1
Alphoma type (α-subunit positive only)	3
Null cell tumor/ultrastructurally gonadotroph	7
Tumors with oncocytic features	17
Ki67 index > 3%	5
**Clinical classification**	
Invasive PitNET	54
Noninvasive PitNET	26
Tumor size > 10 mm	66
Tumor size < 10 mm	14
Newly diagnosed	68
Recurrent	12
Recurrence during follow-up	10

Abbreviations: PitNET—pituitary neuroendocrine tumor; TSH—thyroid-stimulating hormone; FSH—follicle-stimulating hormone; LH—luteinizing hormone.

**Table 2 jcm-10-00560-t002:** The results of assessment of immunoreactivity against L1-ORF1p in nonfunctioning gonadotroph PitNETs.

	Weak Expression	Moderate Expression	High Expression
**Normal pituitary**	2	1	0
**Invasive tumors (*n* = 17)**	0	16	18
**Noninvasive tumors (*n* = 13)**	0	7	12

## Data Availability

Data are presented in main text and Appendix A.

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
