# Peer review of "Invasive and Noninvasive Nonfunctioning Gonadotroph Pituitary Tumors Differ in DNA Methylation Level of LINE-1 Repetitive Elements"

_jcm, 2021, doi:10.3390/jcm10040560_

Round 1
Reviewer 1 Report
Although the authors have adequately addressed most of the comments, there are some minors concerns that should be attended:
- The paragraph “LINE-1 sequence contains two open reading frames, L1-ORF1 and L1-ORF2, that encode for ORF1p (RNA binding protein) and ORF2p (reverse transcriptase and endonuclease) proteins, respectively. These two proteins cooperate to introduce retrotransposon sequence in new chromatin target sites [13]” was not moved from the discussion to the introduction section. This concern was not addressed by the authors. Indeed, although they have improved the introduction section, there is still no information about the two open reading frames.
- The sentence “When we compared expression levels of L1-ORF1 and L1-ORF2 between invasive and noninvasive tumors no significant difference was observed (Figure 2a)” should be moved just before the sentence “Analysis of LINE-1 methylation and expression levels….”. Please also indicate (Figure 2b) at the end of the that sentence “Analysis of LINE-1 methylation and expression levels….”.
- In line 223 should appear Figure 2c instead of Figure2b.
- In the legend of Figure 2, (b) is not indicated.
- No images or graphs are in Figure 3 when the file is downloaded. Please correct this error.
Author Response
Reviewer’s comment: The paragraph “LINE-1 sequence contains two open reading frames, L1-ORF1 and L1-ORF2, that encode for ORF1p (RNA binding protein) and ORF2p (reverse transcriptase and endonuclease) proteins, respectively. These two proteins cooperate to introduce retrotransposon sequence in new chromatin target sites [13]” was not moved from the discussion to the introduction section. This concern was not addressed by the authors. Indeed, although they have improved the introduction section, there is still no information about the two open reading frames.
Reply: We apologies for this omission. The sentence was moved in revised manuscript. We agree it should be placed in introduction section.
Reviewer’s comment: The sentence “When we compared expression levels of L1-ORF1 and L1-ORF2 between invasive and noninvasive tumors no significant difference was observed (Figure 2a)” should be moved just before the sentence “Analysis of LINE-1 methylation and expression levels….”. Please also indicate (Figure 2b) at the end of the that sentence “Analysis of LINE-1 methylation and expression levels….”.
Reply: This was corrected in the revised manuscript.
Reviewer’s comment: In line 223 should appear Figure 2c instead of Figure2b.
Reply: This mistake was corrected in the revised manuscript.
Reviewer’s comment: In the legend of Figure 2, (b) is not indicated.
Reply: This was corrected in the revised manuscript.
Reviewer’s comment: No images or graphs are in Figure 3 when the file is downloaded. Please correct this error.
Reply: The Figure 3 was embedded in revised manuscript MS Word file, as we can easily verify through submission system. It was probably unavailable for the reviewers due to a technical error. This file contains high resolution pictures that may cause a problem with processing the manuscript file. We will pay a special attention to make sure that complete manuscript is provided for the reviewers with this submission.
Reviewer 2 Report
Rusetska et al. analyzed the methylation levels and attempted to evaluate the role of LINE-1 repetitive elements in the tumor growth behavior in a cohort consisting of 54 invasive and 26 noninvasive gonadotroph adenomas. They found that LINE-1 repetitive element was significantly more hypomethylated in nonfunctioning gonadotroph pituitary tumors compared to the non-neoplastic pituitary and in invasive tumors in relation to the noninvasive counterparts. However, neither the expression of two LINE-1 transcripts (by RT-PCR), L1-ORF1 and L1-ORF2 or the protein, L1ORF1p, used as a surrogate for LINE-1 expression, differentiated the growth status of PitNET, most probably reflecting the variability of the evaluated variables. They also observed a significant but low correlation levels between LINE-1 methylation and the expression of the two transcripts but not with the protein. They also found that DNA promoter methylation inhibition increased the expression of both L1-ORF1 and L1-ORF2 transcripts in gonadotroph cell lines despite the response to treatment varied with the cell line type. In conclusion, the manuscript is well-written, the finding of hypomethylation of LINE-1 in pituitary tumors is novel, and despite the authors’ attempts, its functional and prognostic roles remained elusive possibly as a reflection of the small number of samples in each group, the variability of the molecular findings and clinical features of the cohort (primary vs recurrent, oncocytic variants, size etc).
Other comments:
Figure 3 is missing.
Line 52 - in human cancer both local and global DNA methylation changes occur in the genome [9].
Line 179 – add space between from and 75.05%
Supplementary table –
- Column G – misspelling of tumor;
- Provide more details about the Knosp classification, state the difference of primary and newly diagnosed tumors;
- add the values of the molecular findings
Comparison between gonadotroph adenomas with or without oncocytic features?
Discuss the rationale of a "clear relationship between DNA methylation and LINE-1 expression was not observed in LβT2" compared to the result in the other cell line.
Author Response
Reviewer’s comment: Figure 3 is missing.
Reply: The Figure 3 was embedded in revised manuscript MS Word file, as we can easily verify through submission system. It was probably unavailable for the reviewers due to a technical error. This file contains high resolution pictures that may cause a problem with processing the manuscript file. We will pay a special attention to make sure that complete manuscript is provided for the reviewers with this submission.
Reviewer’s comment: Line 52 - in human cancer both local and global DNA methylation changes occur in the genome [9].
Reply: Corrected according to the suggestion.
Reviewer’s comment: 179 – add space between from and 75.05%
Reply: Corrected
Reviewer’s comment: Supplementary table – Column G – misspelling of tumor;
Reply: Corrected in the revised supplementary Table.
Reviewer’s comment: Provide more details about the Knosp classification, state the difference of primary and newly diagnosed tumors;
add the values of the molecular findings
Reply: We introduced the column with details of Knosp classification for each patient.
We mistakenly used of both terms “primary” and “newly diagnosed”. It is the same category. “Primary” was used in the meaning of newly diagnosed, opposite to recurrent tumors. In the revised supplementary table we used “newly diagnosed” as category.
The results of molecular analysis (L1 DNA methylation level and relative L1-ORF1 and L1-ORF2 expression) for each patient were introduced in revised supplementary table 1.
Reviewer’s comment: Comparison between gonadotroph adenomas with or without oncocytic features?
Reply: No significant difference in LINE-1 methylation was observed between oncocytomas and tumors without oncocytic features. This information was introduced in the revised manuscript in the results section 3.1.
Reviewer’s comment: Discuss the rationale of a "clear relationship between DNA methylation and LINE-1 expression was not observed in LβT2" compared to the result in the other cell line.
Reply: It appears that αT3-1 cells are more sensitive to DNA methylation inhibitor. Both cell lines were derived from normal mouse gonadotropic pituitary cells (not tumor cells), however, the main difference between these cells is that they represent gonadotroph cells at different stages of differentiation. αT3-1 cells were derived by immortalization of mouse pituitary gonadotroph cells at the stage of expression od alpha-subunit, but before activation of LHB gene that encodes LH (beta subunit). Thus they are immature gonadotroph cells. In turn, LβT2 cells were immortalized at the stage of expression both alpha and LH-beta subunits, therefore they represent mature gonadotrophs.
Epigenetic regulation plays a role in differentiation of pituitary cells [1–3]. Epigenetic regulation of alternative promoters and enhancers in progenitor, immature, and mature gonadotrope cell lines; Chromatin status and transcription factor binding to gonadotropin promoters in gonadotrope cell lines) and it can be expected that immature and mature cells of the same lineage may differ in terms of regulation of epigenetic machinery.
This includes difference in editing DNA methylation pattern. It was clearly shown that αT3-1 cells have higher expression level of Tet1 protein which is directly involved in active DNA demethylation [3]. It is possible that due to physiological DNA demethylation activity is higher in αT3-1, experimental inhibition of DNA methylatrsferases is more effective in these cells than in LβT2 cells (with lower DNA demethylation activity). Both natural active demethylation process and inhibiting DNA methyltransferases act synergistically.
Of course this explanation is of speculative nature. This explanation was introduced in Discussion section in the revised manuscript.
- Laverrière, J.N.; L’Hôte, D.; Tabouy, L.; Schang, A.L.; Quérat, B.; Cohen-Tannoudji, J. Epigenetic regulation of alternative promoters and enhancers in progenitor, immature, and mature gonadotrope cell lines. Mol. Cell. Endocrinol. 2016, 434, 250–265.
- Xie, H.; Hoffmann, H.M.; Iyer, A.K.; Brayman, M.J.; Ngo, C.; Sunshine, M.J.; Mellon, P.L. Chromatin status and transcription factor binding to gonadotropin promoters in gonadotrope cell lines. Reprod. Biol. Endocrinol. 2017, 15, 86.
- Yosefzon, Y.; David, C.; Tsukerman, A.; Pnueli, L.; Qiao, S.; Boehm, U.; Melamed, P. An epigenetic switch repressing Tet1 in gonadotropes activates the reproductive axis. Proc. Natl. Acad. Sci. 2017, 114, 201704393.
Round 2
Reviewer 2 Report
- Small typos are still present in the table ("recurent", "Charactereistics")
- Unit of LINE-1 methylation levels is missing
Author Response
Reviewer’s comment: Small typos are still present in the table ("recurent", "Charactereistics")
Reply: These typos in Supplementary table were corrected. The entire manuscript was carefully checked for the errors and additional typographical and grammatical errors were found and corrected.
And Figure 1 was changed due to a typographical error in Figure 1c
Reviewer’s comment: Unit of LINE-1 methylation levels is missing
Reply: The unit (%) was added at each value in column with DNA methylation level data in Supplementary table.
This manuscript is a resubmission of an earlier submission. The following is a list of the peer review reports and author responses from that submission.
Round 1
Reviewer 1 Report
I really liked the scientific design of the manuscript. The methods are well presented and the results answer the scientific question.
My suggestion for the manuscript: Invasive and noninvasive nonfunctioning pituitary tumors differ in DNA methylation level of LINE-1 repetitive elements
In this article the authors emphasize the importance of global DNA methylation levels in PiTNETs.
The data are well presented, but they are clearly preliminary data, given the limited numbers. It could be presented as a pilot study.
A suggestion could be the evaluation of tumor aggressiveness with evaluation of proliferation factors and tumor suppressor genes and possibly the number of mitoses present.
Thanks
Author Response
Reviewer 1
My suggestion for the manuscript: Invasive and noninvasive nonfunctioning pituitary tumors differ in DNA methylation level of LINE-1 repetitive elements
In this article the authors emphasize the importance of global DNA methylation levels in PiTNETs.
The data are well presented, but they are clearly preliminary data, given the limited numbers. It could be presented as a pilot study.
Reviewer’s comment: A suggestion could be the evaluation of tumor aggressiveness with evaluation of proliferation factors and tumor suppressor genes and possibly the number of mitoses present.
Reply: This issue was also raised by the other reviewers. In the revised manuscript we used results of ki67 staining and data regarding recurrence to investigate whether difference LINE-1 methylation can be found in patients stratified according to these parameters – whether it plays a role in aggressive tumor growth. We didn’t find difference between patients with ki67 <3% and >3% (the major criterion of atypical pituitary tumor). No differences between primary and recurrent tumors as well as tumors that recurred within follow-up and those without relapse were also found. Unfortunately, low numbers of patients could be used for these analyses since only 5 patients had ki67index >3%; 12 recurrent tumors were enrolled in the study group and recurrence in follow up was observed in only 10 patients. Probably a longer follow-up (in this study median follow-up is 5 years 10 month (71 months)) would allow collecting more valuable data on tumor recurrence after treatment. In revised manuscript the results are described in results section 3.1. Clinical data on ki67 staining were included in Table 1 with sample description and in Supplementary Table 1.
Reviewer 2 Report
This report examines the methylation level of Long Interspersed Nuclear Element-1 (LINE-1) repetitive elements and the expression of two open reading frames (L1-ORF1 and L1-ORF2) contained in LINE-1 sequence in invasive and noninvasive nonfunctioning PitNETs.General concerns:
- The authors use either gonadotroph PitNETs or nonfunctioning tumors along the manuscript. This nomenclature is confusing because gonadotroph PitNETs can be functioning or nonfunctioning PitNETs. Since they explained in the introduction that the majority of nonfunctioning PitNETs have a gonadotropic origin, they should use nonfunctioning PitNETs in all the manuscript to be more consistent.
- Reduced LINE-1 methylation has been reported in many tumor types. The authors describe a reduction in LINE-1 methylation in invasive vs noninvasive nonfunctioning PitNETs. However, they do not show LINE-1 methylation levels in normal pituitary glands, which is absolutely necessary to understand the role of LINE-1 methylation in nonfunctioning PitNETs.
- Regarding the experiments in cell lines, the authors show a significant increase in L1-ORF1 and L1-ORF2 expression levels after 5-AzaC treatment in aT3-1 but not in LbT2 cells. First, are the significant p-value indicated in the graphs related to the highest 5-AzaC concentration? Secondly, they only show two independent experiments (n=2) which is not sufficient to clearly distinguish the real effect of 5-AzaC and the relationship of DNA methylation and L1-ORF1 and L1-ORF2 expression levels, neither to perform a statistical analysis. The number of experiments should be increased. Moreover, wound healing or transwell invasion experiments after 5-AzaC treatment should be done to understand if the reduced methylation and the increase in L1-ORF1 and L1-ORF2 expression levels are related with an increase in invasiveness.
- The authors also show immunohistochemistry results of L1-ORF1p in invasive and noninvasive tumors using an antibody previously reported by Rodíc et al. 2014. They show a moderate to high expression in all tumors analyzed with a really homogeneous stain, but no positive or negative controls are included to ensure that the staining is completely specific. Rodíc et al. report the expression of L1-ORF1p in different tissues that can be used as internal controls. Additionally, normal pituitary glands should be also analyzed.
- The abstract and introduction sections should be improved. Lines 243-246 from discussion should be moved to introduction section.
- Is there a correlation between L1-ORF1p and clinical/pathological parameters like Ki67 or p53?
Minor concerns:
- Line 124: 5-Azac should appear with the complete name since is the first time that is described.
- In cell culture section should be indicated if the cell lines have been tested for mycoplasma contamination.
- Section 3.1: it is described that LINE-1 methylation levels were determined in 54 invasive and 26 noninvasive tumors, which do not match with the numbers that appear in table 1.
- Figure 1a: it is indicated “invasive and noninvasive NFPA” but NFPA nomenclature has not be mentioned previously.
- Section 3.3: the authors describe “Analysis of LINE-1 methylation and expression levels showed significant negative correlation for both mRNAs: L1-ORF1 (Spearman R=0.2680; p=0.0169) and L1-ORF2 (Spearman R=0.2720; p=0.0153)”. The graph of this correlation has to be included in figure 2.
- Line 179: the p-values described regarding L1-ORF1 and L1-ORF2 expression levels don´t match with the p-values that appear in the graphs in figure 2a, and is indicated “Figure 1B” instead of Figure 2A.
- Lines 228-230: include references to support the sentence.
Author Response
Reviewer 2
This report examines the methylation level of Long Interspersed Nuclear Element-1 (LINE-1) repetitive elements and the expression of two open reading frames (L1-ORF1 and L1-ORF2) contained in LINE-1 sequence in invasive and noninvasive nonfunctioning PitNETs.
General concerns:
Reviewer’s comment: The authors use either gonadotroph PitNETs or nonfunctioning tumors along the manuscript. This nomenclature is confusing because gonadotroph PitNETs can be functioning or nonfunctioning PitNETs. Since they explained in the introduction that the majority of nonfunctioning PitNETs have a gonadotropic origin, they should use nonfunctioning PitNETs in all the manuscript to be more consistent.
Reply: This was corrected, we used term nonfunctioning gonadotroph PitNETs along the manuscript. We also introduced the statement “All the samples were nonfunctioning gonadotroph PitNETs.” in section 2.1 Patients and samples to clearly indicate that no functioning gonadotroph tumors were included.
Reviewer’s comment: Reduced LINE-1 methylation has been reported in many tumor types. The authors describe a reduction in LINE-1 methylation in invasive vs noninvasive nonfunctioning PitNETs. However, they do not show LINE-1 methylation levels in normal pituitary glands, which is absolutely necessary to understand the role of LINE-1 methylation in nonfunctioning PitNETs.
Reply: We introduced DNA methylation results for normal pituitary in the revised manuscript. This shows a higher DNA methylation level at LINE-1 elements in normal pituitary gland. Samples of normal pituitary were also used for immunostaining with anti L1-ORF1p antibody.
In section 2.1 and supplementary Table 1 we added information on normal pituitary samples. The results of the investigation of normal pituitary samples were described in sections “3.1 Results” and are presented in Figures 1a; 2a and 3. The results of normal tissue analysis were also commented in discussion section.
Reviewer’s comment: Regarding the experiments in cell lines, the authors show a significant increase in L1-ORF1 and L1-ORF2 expression levels after 5-AzaC treatment in aT3-1 but not in LbT2 cells. First, are the significant p-value indicated in the graphs related to the highest 5-AzaC concentration? Secondly, they only show two independent experiments (n=2) which is not sufficient to clearly distinguish the real effect of 5-AzaC and the relationship of DNA methylation and L1-ORF1 and L1-ORF2 expression levels, neither to perform a statistical analysis. The number of experiments should be increased. Moreover, wound healing or transwell invasion experiments after 5-AzaC treatment should be done to understand if the reduced methylation and the increase in L1-ORF1 and L1-ORF2 expression levels are related with an increase in invasiveness.
Reply. We performed two additional biological replicates of experiment of measuring L1-ORF1 and L1-ORF2 expression levels in cells treated with different doses of 5-AzaC. In the revised manuscript the results of four replicates are described in results section 3.3 and presented in Figure 2 c. The revised graph shows each replicated experiment. Data were analyzed with Friedman test which is nonparametric equivalent of ANOVA test for comparing more than two groups of samples.
In general, 5-AzaC has a strong cytotoxic effect on most of the cells. This agent is used as cytostatic drug in hematological neoplasms. 5-AzaC negatively influences the growth of both gonadotroph cell lines as indicated by notably lower cell numbers and lower metabolic activity (MTT test results, data not shown) in cultures with increasing concentrations of this methyltransferase inhibitor. We believe that performing any functional test on the cells treated with 5-AzaC would show the direct cytotoxic effect and would not reflect the indirect effect of increased expression of L1 elements. For this reason these functional tests were not performed.
Reviewer’s comment: The authors also show immunohistochemistry results of L1-ORF1p in invasive and noninvasive tumors using an antibody previously reported by Rodíc et al. 2014. They show a moderate to high expression in all tumors analyzed with a really homogeneous stain, but no positive or negative controls are included to ensure that the staining is completely specific. Rodíc et al. report the expression of L1-ORF1p in different tissues that can be used as internal controls. Additionally, normal pituitary glands should be also analyzed.
Reply: In the revised manuscript we included pictures of controls used for immunoreactivity staining: colorectal adenocarcinoma sample – positive control; normal kidney – negative control (both selected according to previous reports (Harris et al., 2010; Rodić et al., 2014) and negative control (without primary antibody). We used three samples of FFPE normal pituitary tissue for staining with ati-L1orf1p antibody. Two of these normal samples showed weak immunoreactivity clearly lower than observed in pituitary tumor samples. One normal pituitary section showed moderate immunoreactivity. Immunostaining was also performed on additional 23 tumor samples. For more accurate analysis staining reactivity was quantified using H-score formula (McCarty et al., 1986) that combines information on number of positive cells and staining intensity. Results of staining normal pituitary tissue and control samples as well as results of H-score comparison were described in revised results section 3.4 and are shown in revised Figure 3.
Reviewer’s comment: The abstract and introduction sections should be improved. Lines 243-246 from discussion should be moved to introduction section.
Reply: We moved the mentioned lines as suggested by the reviewer and made an attempt to improve introduction section.
Reviewer’s comment: Is there a correlation between L1-ORF1p and clinical/pathological parameters like Ki67 or p53?
We checked whether there is a relationship between Ki67 and L1-ORF1p by comparing H-score of tumors with low Ki67 <3% and Ki67>3%. This threshold was used since it was basic criterion for atypical pituitary tumor and this classification was available in patients records. We observed no difference, however, we have to note that only 5 patients had ki67 >3%. Staining results against TP53 was unavailable for large proportion of patients and therefore it was not analyzed. We also tested whether there is difference in H-score in newly diagnosed (n=68) and recurrent tumors (n=12) and we observed the lack of difference. We decided not to include these results in the manuscript for two reasons:
- the number of patients with proliferation index >3% as well as patients with recurrent tumors are small (5 and 12 samples, respectively), so the quality of the analysis is rather poor.
- In LINE-1 DNA methylation analysis only comparison of invasive vs noninvasive PitNETs showed significant difference. We believe this is the result that should be complemented by subsequent expression analysis. No difference was observed in patients stratified according to Ki67 labeling or patients with newly diagnosed vs recurrent tumors.
Minor concerns:
Reviewer’s comment: Line 124: 5-Azac should appear with the complete name since is the first time that is described.
Reply: Corrected in revised manuscript.
Reviewer’s comment: In cell culture section should be indicated if the cell lines have been tested for mycoplasma contamination.
Reply: Cells were determined as mycoplasma free using PCR assay. This information was added to methods section 2.6
Reviewer’s comment: Section 3.1: it is described that LINE-1 methylation levels were determined in 54 invasive and 26 noninvasive tumors, which do not match with the numbers that appear in table 1.
Reply: The mistake was corrected. 54 invasive and 26 noninvasive tumors were included.
Reviewer’s comment: Figure 1a: it is indicated “invasive and noninvasive NFPA” but NFPA nomenclature has not be mentioned previously.
Reply: We agree this should be corrected. We used terms invasive PitNETs and noninvasive PitNETs in Figures 1 and 2 in the revised manuscript.
Reviewer’s comment: Section 3.3: the authors describe “Analysis of LINE-1 methylation and expression levels showed significant negative correlation for both mRNAs: L1-ORF1 (Spearman R=0.2680; p=0.0169) and L1-ORF2 (Spearman R=0.2720; p=0.0153)”. The graph of this correlation has to be included in figure 2.
Reply: We introduced correlation graphs as Figure 2b in the revised manuscript
Reviewer’s comment: Line 179: the p-values described regarding L1-ORF1 and L1-ORF2 expression levels don´t match with the p-values that appear in the graphs in figure 2a, and is indicated “Figure 1B” instead of Figure 2A.
Reply: Following suggestion from Reviewer 3 this sentence was removed because no significant difference was observed. We verified that p-values shown in Figure 2 are correct.
Reviewer’s comment: Lines 228-230: include references to support the sentence.
Reply: The reference was provided in revised manuscript
arris, C. R. et al. (2010) ‘Association of nuclear localization of a long interspersed nuclear element-1 protein in breast tumors with poor prognostic outcomes’, Genes and Cancer, 1(2), pp. 115–124. doi: 10.1177/1947601909360812.
McCarty, K. S. et al. (1986) ‘Use of a monoclonal anti-estrogen receptor antibody in the immunohistochemical evaluation of human tumors’, Cancer Research, 46(8 SUPPL.), pp. 4244–4249.
Nordlund, J. et al. (2013) ‘Genome-wide signatures of differential DNA methylation in pediatric acute lymphoblastic leukemia’, Genome Biology, 14(9), p. r105. doi: 10.1186/gb-2013-14-9-r105.
Rodić, N. et al. (2014) ‘Long interspersed element-1 protein expression is a hallmark of many human cancers’, American Journal of Pathology, 184(5), pp. 1280–1286. doi: 10.1016/j.ajpath.2014.01.007.
Sur, D. et al. (2017) ‘Detection of the LINE-1 retrotransposon RNA-binding protein ORF1p in different anatomical regions of the human brain’, Mobile DNA. Mobile DNA, 8(1), pp. 1–12. doi: 10.1186/s13100-017-0101-4.
Reviewer 3 Report
The results of study are not very novel since global DNA methylation levels have been previously analyzed in PiTNETs (including by the authors themselves). Nevertheless, in general the studies have been correctly performed although some issues need to be addressed.
Specific comments:
1. Table 1 is not very intuitive to read. Reorganize, there is no need to have median and ranges in different files.
2. Include tumor size median in Table 1.
3. The authors describe in the introduction that decreased methylation of repetitive DNA sequences is a surrogate marker for global DNA hypomethylation. Wouldn’t the authors expect the correlation shown in Figure 1c to be stronger?
4. Invasion might be correlated with tumor size. The authors should determine whether LINE-1 DNA methylation levels correlate with tumor size. More importantly, have the authors evaluated whether LINE-1 DNA methylation levels correlate with other aggressive features of PitNETs such as regrowth/recurrence or high Ki67 levels? These studies could help to support the authors hypothesis that lower LINE-1 methylation are relative with aggressive tumor growth.
5. Line 177-179 “but the difference was not statistically significant”. If there not statistical significance what is the point of indicating that “a higher expression level of both transcripts was observed in invasive PitNETs”? Please, rephrase.
6. Line 179. Figure1b is mistakenly referenced. I assume they meant Figure 2a: If so, the p values in the text don’t match the p values shown in the figures.
7. Why only 30 tumors (out of the 80 patients included in the study) were analyzed by immunohistochemical staining? Is there any possibility of bias selection? How do the authors explain there is no correlation between L1-ORF1p immunoreactivity and LINE-1 DNA methylation or L1 transcripts expression levels?
8. Provide higher magnification (and better) pictures for Figure 3 so the expression pattern can be easily observed.
9. Discussion, line 260: a slightly higher expression was detected. There is no statistical significance so please remove this statement.
Author Response
Reviewer 3
The results of study are not very novel since global DNA methylation levels have been previously analyzed in PiTNETs (including by the authors themselves). Nevertheless, in general the studies have been correctly performed although some issues need to be addressed.
Specific comments:
Reviewer’s comment: Table 1 is not very intuitive to read. Reorganize, there is no need to have median and ranges in different files.
Reply: We made an attempt improve Table 1 by separation of demographical and clinical data.
Reviewer’s comment: Include tumor size median in Table 1.
Reply: Unfortunately we don’t have the exact tumor size data in many of patients records. The classification of macroadenoma (diameter >10mm ) vs microadenoma (diameter <10 mm ) was available and introduced in Table 1 and Supplementary Table 1. We were unable to provide median of tumor size.
Reviewer’s comment: The authors describe in the introduction that decreased methylation of repetitive DNA sequences is a surrogate marker for global DNA hypomethylation. Wouldn’t the authors expect the correlation shown in Figure 1c to be stronger?
Reply: Figure 1c shows the correlation between LINE-1 methylation and overall methylation level calculated from HM450K methylation arrays. The correlation with Spearman R=0.58 was observed. Of course we would like to see a stronger correlation. We believe that this result is due to the fact that most of HM450 array probes are concentrated at genes and gene regulatory regions and therefore the generalized results of methylation data from H450 arrays reflect the median of methylation at regions covered by the array probes but not true whole genome methylation. Signal from the probes aligning to multiple genomic locations are, a priori, removed from analysis, according to recommendations (Nordlund et al., 2013)and no probes for repetitive elements are used for data normalization and analysis. In our manuscript we used the term “global genome-wide methylation assessed with HM450K methylation arrays” and intentionally avoid term “whole genome methylation”. In discussion section of the revised manuscript we introduced the statement that “The relationship that we observed is not strong but it should be noted that genome-wide methylation assessed with methylation arrays may not reflect the true genomic methylation level but rather overall methylation status of genomic regions covered by array probes.”
Reviewer’s comment: Invasion might be correlated with tumor size. The authors should determine whether LINE-1 DNA methylation levels correlate with tumor size. More importantly, have the authors evaluated whether LINE-1 DNA methylation levels correlate with other aggressive features of PitNETs such as regrowth/recurrence or high Ki67 levels? These studies could help to support the authors hypothesis that lower LINE-1 methylation are relative with aggressive tumor growth.
Reply. According to reply to previous comment we didn’t have exact data of tumor size on classification of macroadenoma vs microadenoma. No difference between macroadenomas and microadenomas were observed. Unfortunately, vast majority of tumors in the study group are macroadenomas (n=66). This bias results from tissue collecting. Commonly neurosurgeons don’t have possibility to collect and freeze small microadenoma tissue samples for research because they have to preserve tissue for diagnosis.
In the revised manuscript we used results of ki67 staining and data regarding recurrence to investigate whether difference in LINE-1 methylation can be found in patients stratified according to these parameters – whether it plays a role in aggressive tumor growth. We didn’t find difference between patients with ki67 <3% and >3% (the major criterion of atypical pituitary tumor). No difference between primary and recurrent tumors as well as tumors that recurred within follow-up and those without relapse were also found. Unfortunately, low numbers of patients could be used for these analyses since only 5 patients had ki67 >3%; 12 recurrent tumors were enrolled in study group and recurrence in follow-up was observed in only 10 patients. Additionally, for more valuable analysis of tumor recurrence we would rather need longer follow-up to collect data on tumor recurrence after treatment. For our cohort median follow-up is 5 years 10 month (71 months)). In the revised manuscript the results are described in results section 3.1. Clinical data on ki67 staining and recurrence status were included in Table 1 and in Supplementary Table 1.
Reviewer’s comment: Line 177-179 “but the difference was not statistically significant”. If there not statistical significance what is the point of indicating that “a higher expression level of both transcripts was observed in invasive PitNETs”? Please, rephrase.
Reply: This sentence was corrected: “When we compared expression levels of L1-ORF1 and L1-ORF2 between invasive and noninvasive tumors no significant difference was observed”
Reviewer’s comment. Line 179. Figure1b is mistakenly referenced. I assume they meant Figure 2a: If so, the p values in the text don’t match the p values shown in the figures.
Reply: The mistake has been corrected as p-values provided in line 179 were removed from the text following the previous suggestion. We verified that p-values in Figure 2a are correct.
Reviewer’s comment: Why only 30 tumors (out of the 80 patients included in the study) were analyzed by immunohistochemical staining? Is there any possibility of bias selection? How do the authors explain there is no correlation between L1-ORF1p immunoreactivity and LINE-1 DNA methylation or L1 transcripts expression levels?
Reply: The relationship between L1-ORF1p expression and LINE-1 DNA methylation was more carefully investigated for manuscript revising. We performed immunostaining on additional 23 tumor samples. Staining reactivity was quantified using H-score formula (McCarty et al., 1986)that combines information on number of positive cells and staining intensity. This generated numerical data suitable for correlation analysis. We made correlation analysis of L1-ORF1p expression and methylation level and we didn’t observe correlation. We suspect that two things may be the cause of lack of the correlation in our results: difficulties in quantification of immunohistochemical data which always rely on subjective observation and natural differences in expression levels between individual patients, that are not a result of impaired methylation but reflect normal inter-individual diversity. Such differences were observed in previous examination of normal human brain tissues (Sur et al., 2017) and we think they might be also the cause of different immunoreactivity of one of three normal pituitary samples that we show in the revised manuscript. This comment was introduced in discussion section of the revised manuscript.
Reviewer’s comment: Provide higher magnification (and better) pictures for Figure 3 so the expression pattern can be easily observed.
Reply: We replaced pictures in Figure 3 using pictures with x400 magnification
Reviewer’s comment: Discussion, line 260: a slightly higher expression was detected. There is no statistical significance so please remove this statement.
Reply: The statement was removed.
Harris, C. R. et al. (2010) ‘Association of nuclear localization of a long interspersed nuclear element-1 protein in breast tumors with poor prognostic outcomes’, Genes and Cancer, 1(2), pp. 115–124. doi: 10.1177/1947601909360812.
McCarty, K. S. et al. (1986) ‘Use of a monoclonal anti-estrogen receptor antibody in the immunohistochemical evaluation of human tumors’, Cancer Research, 46(8 SUPPL.), pp. 4244–4249.
Nordlund, J. et al. (2013) ‘Genome-wide signatures of differential DNA methylation in pediatric acute lymphoblastic leukemia’, Genome Biology, 14(9), p. r105. doi: 10.1186/gb-2013-14-9-r105.
Rodić, N. et al. (2014) ‘Long interspersed element-1 protein expression is a hallmark of many human cancers’, American Journal of Pathology, 184(5), pp. 1280–1286. doi: 10.1016/j.ajpath.2014.01.007.
Sur, D. et al. (2017) ‘Detection of the LINE-1 retrotransposon RNA-binding protein ORF1p in different anatomical regions of the human brain’, Mobile DNA. Mobile DNA, 8(1), pp. 1–12. doi: 10.1186/s13100-017-0101-4.